# A Systematic Review of Pharmacological Interventions for Apathy in Aging Neurocognitive Disorders

**DOI:** 10.3390/brainsci13071061

**Published:** 2023-07-12

**Authors:** Christos Theleritis, Kostas Siarkos, Anastasios Politis, Nikolaos Smyrnis, Charalabos Papageorgiou, Antonios M. Politis

**Affiliations:** 1First Department of Psychiatry, National and Kapodistrian University of Athens, Eginition Hospital, 74 Vas. Sofias Ave., 11528 Athens, Greece; chpapag@med.uoa.gr (C.P.); apolitis@med.uoa.gr (A.M.P.); 2Second Department of Neurosurgery, National and Kapodistrian University of Athens, Attikon Hospital, 1 Rimini Str., 12462 Athens, Greece; anastasios.politis@nhs.net; 3Second Department of Psychiatry, National and Kapodistrian University of Athens, Attikon Hospital, 1 Rimini Str., 12462 Athens, Greece; smyrnis@med.uoa.gr; 4Department of Psychiatry and Behavioral Sciences, Johns Hopkins University, Baltimore, MD 21218, USA

**Keywords:** apathy, aging neurocognitive disorders, dementia, Alzheimer’s disease (AD), Parkinson’s disease (PD), Huntington’s disease (HD), frontotemporal dementia (FTD), pharmacological treatments

## Abstract

Objective: Apathy, a frequent neuropsychiatric symptom in aging neurocognitive disorders, has been associated with cognitive decline and functional disability. Therefore, timely provision of pharmacological interventions for apathy is greatly needed. Design: A systematical literature review of existing studies was conducted up to 30 May 2023 in several databases (PubMed, PsychInfo, Cochrane, Google Scholar, etc.) that included randomized controlled trials (RCTs) and meta-analyses assessing pharmacological treatments for apathy in aging neurocognitive disorders. The quality of the studies was appraised. Results: In patients with Alzheimer’s Disease (AD), donepezil, galantamine, rivastigmine, methylphenidate, and gingko biloba were proven efficacious for apathy, while rivastigmine, cognitive enhancer IRL752 and piribedil were found to be beneficial in patients with Parkinson’s Disease (PD) and agomelatine in patients with Frontotemporal Dementia (FD). The extensive proportion of RCTs in which apathy was used as a secondary outcome measure, along with the considerable methodological heterogeneity, did not allow the evaluation of group effects. Conclusions: Pharmacological interventions for apathy in aging neurocognitive disorders are complex and under-investigated. The continuation of systematic research efforts and the provision of individualized treatment for patients suffering from these disorders is vital.

## 1. Introduction

Apathy is encountered in several neuropsychiatric disorders; it is present in 27% to 72% of patients with AD [1,2,3,4,5], up to 90% of patients with Frontotemporal Dementia (FTD), dementia with Lewy bodies (DLB) and progressive supranuclear palsy, in 40% of those with cortico-basal degeneration, and in 20% of those with Parkinson’s disease (PD) [2,6]. Some degree of apathy is observed in brain injuries and cerebrovascular lesions concerning frontal lobes and is related largely to lesion location [2,7].

In patients experiencing apathy, frontal-subcortical brain areas seem to be dysfunctional; this fact undermines the efficient selection, initiation, maintenance and changing of action programs from the frontal cortex [8,9,10,11]. 

Various definitions for apathy have been proposed over the years [7,12,13,14]. Recently, the following criteria [15] for patients with apathy in neurocognitive disorders (NCD) were proposed: symptoms persistent or frequently recurrent over at least 4 weeks, a change from the patient’s usual behavior, and including one of the following: diminished initiative, diminished interest, or diminished emotional expression/responsiveness; causing significant functional impairment and not exclusively explained by other etiologies. 

Apathy has been associated with cognitive decline and functional disability and is often encountered in patients with mild cognitive impairment (MCI), dementia, and other neurocognitive disorders both in nursing homes and community samples [5,16,17,18,19]. Apathy increases the risk of developing AD [20], it is increasingly persistent [21], accelerates cognitive decline [21], drug-resisting, pervasive, and silent, disabling, causing hospitalization, mortal, affecting the quality of life, increases the risk of caregiver distress [22]. Different aspects of apathy are described, such as cognitive, emotional, and social apathy. Importantly, apathy is present in a series of conditions, such as psychiatric disorders (schizophrenia, depression), or as a neuropsychiatric symptom in neurodegenerative states, brain trauma, and stroke. Apathy, as a symptom, may manifest differently with respect to the underlying disorder, while in healthy populations, subclinical apathy in women is associated with changes in brain pathways known to be implicated in clinical apathy [23]. Stuss et al. [24] theorize apathy as a distinct but related clinical state related to the neural substrate and/or the behavioral response involved.

To summarize, apathy is among the BPSD states that are highly prevalent in several neuropsychiatric disorders (AD, FTD, DLB, PD), affecting from 20% to 90% of the patients with these conditions, depending upon the type of impairment. Although the etiology of apathy is not yet fully understood and may differ between impairments, there are nonetheless many efforts underway to find pharmacologic interventions for apathy—or mitigate its development and/or magnitude. In previous reviews [25,26,27], we searched for effectiveness in apathy throughout treatment modalities. As of yet, however, no standard or approved approach has emerged. Therefore, this paper strives to systematically review and describe the current state of knowledge in the field about pharmacological treatments for apathy in these conditions. It expands on pharmacological treatments for apathy in all-cause aging neurocognitive disorders in an effort to address methodological issues, inform practices and appropriately guide future research.

### 1.1. Eligibility Criteria, Information Sources, and Search Strategy

We conducted a systematical literature review of existing randomized controlled trials (RCTs) and meta-analyses in several databases (PubMed, PsychInfo, Cochrane, Google Scholar, etc.) for “apathy and neurocognitive disorders” (last search was on 30 May 2023, 2385 results). Search terms included apart from apathy: frontotemporal dementia, dementia, dementia with Lewy bodies and progressive supranuclear palsy, Parkinson’s disease, Alzheimer’s disease, Huntington’s disease, methylphenidate, antiparkinsonic drugs, acetylcholinesterase inhibitor, memantine, amantadine, modafinil, rotigotine, rasagiline, atomoxetine, piribedil, oxytocin, dextroamphetamine, antidepressants, agomelatine, atypical antipsychotic, pharmacological.

### 1.2. Study Selection and Quality Evaluation

Thus, we identified trials in which apathetic patients suffering from several aging neurocognitive disorders were diagnosed with a reported outcome measure on apathy and received pharmacological agents in controlled designs. We also investigated the references of published articles. Other neuropsychiatric manifestations, as well as concomitant psychoactive medications, were allowed. Typical antipsychotics were not investigated [28]. Relevant abstracts were identified by three authors. Articles were read in full; there was always a consensus meeting with the last author before including related meta-analyses and RCTs. 

OCEBM Levels of Evidence and Grades of Recommendation were used for the classification of all RCTs [29]. There was a further evaluation of RCTs with the (PEDro) rating scale [30]. To evaluate bias in the RCT studies, we utilized items of the PEDro scale overlapping with those of the Cochrane Collaboration tool [31]. Nevertheless, we followed the structure of a systematic review reporting suggested by the Centre for Reviews and Dissemination and appraised the methodological quality in each study semi-quantitatively using tools that evaluate different domains as potential sources of bias and further assessed the level of evidence using OCEBM rating. Strategies, however, followed to minimize biases were a thorough study search, the use of appropriate criteria for study inclusion/exclusion with no changes in the review protocol, the utilization of standard tools to assess the individual study quality and appraise the level of evidence, while assessing for domains common in the most frequently used risk of bias tools. Two raters performed the assessments independently. For study selection, a great variety of keywords was used to ensure certain studies were not excluded. Finally, assessing for duplicates served to minimize bias due to multiple reporting.

### 1.3. Search Results

A search with the terms “apathy” and “neurocognitive disorders” yielded 2385 results (Figure 1). A search with the terms “apathy” and “dementia” yielded 2316 results. A search with the combined terms “apathy,” “neurocognitive disorders,” and “treatment” produced 1131 results; for “apathy,” “neurocognitive disorders,” “pharmacological treatment” 384 results, of “apathy,” “Lewy Bodies Dementia,” “treatment” 56 results, of “apathy,” “Huntington’s Disease,” “treatment” 75 results, of “apathy,” “Parkinson’s Disease,” “treatment” 575 results, of “apathy,” “Frontotemporal Dementia,” “treatment” 115 results. In the final review, sixty-six RCTs, two meta-analyses, and three pooled data analyses on AcheIs were considered. Based on PEDro scores and OCEBM evidence, most studies were of high quality. Multiple heterogeneity and small effect sizes were present. 

### 1.4. Assessment of Apathy

In many studies reported in the literature, apathy has been assessed using the Neuropsychiatric Inventory (NPI). NPI utilizes a semi-structured interview (administered by a clinician to a caregiver for a patient) to assess 12 behavioral domains, of which apathy is one. The twelve domains include delusions, hallucinations, dysphoria, euphoria, apathy, anxiety, agitation/aggression, irritability/lability, disinhibition, aberrant motor behavior, sleep, and appetite/eating disturbances. For each domain, the presence (yes/no questions) of problematic behavior is first evaluated. The ‘informant/caregiver’ rates the neuropsychiatric symptoms in terms of frequency (1 = rarely, 2 = sometimes, 3 = often, 4 = very often) and severity (1 = mild, 2 = moderate, 3 = severe). To obtain a domain score, the frequency of each behavior is multiplied by the severity rating. 

A total Overall NPI Score is obtained by summing all individual domain scores. Of note, a measure of the level of caregiver distress is also obtained, but it is not included in the NPI total score. The NPI has been evaluated psychometrically in many studies, and has shown good content validity, concurrent validity, inter-rater reliability across behaviors and domains, and acceptable test–retest reliability [32].

The Apathy Scale (AS), which was developed by Marin [33], is also frequently used. Marin’s assessment tool originally featured three subscales, one given by the examiner, one by a related other, and one by the patient, respectively. For patients with cognitive impairments, it can be too demanding to administer a questionnaire to themselves, and thus clinician-administered versions are often used. AS scores range from 0 to 42 [34], and higher scores indicate greater severity. Cutoff scores for classifying patients as apathetic or non-apathetic depend on the version used of 14 points used as the threshold. Patients with a score of 14 or higher were classified as apathetic. Starkstein et al. [34] reported that using a threshold of 14 and lower to classify non-apathy, the sensitivity of AS was 66%, and the specificity was 100%. Furthermore, statistical testing showed that the two groups (apathetic vs. non-apathetic) scored significantly differently on the test. Starkstein et al. [34] reported further that: “the same scale was piloted in patients with AD, HD, and stroke (C. Peyser, M.D., and P. Fedoroff, M.D., personal communication to [34]) and was found to have very high intra- and interrater reliability”.

## 2. Review of Pharmacological Treatments (Table 1)

### 2.1. Alzheimer’s Disease (AD)

#### 2.1.1. Donepezil

In a randomized controlled trial, no significant differences in NPI apathy scores were found between placebo and donepezil-treated patients [35]. On the contrary, two retrospective sub-analyses [36,37] of an RCT [38] exhibited significant effects after donepezil treatment vs. placebo. In another two RCTs, NPI apathy scores [39] improved significantly, and AS scores [40] scored non-significantly higher, respectively, with donepezil treatment. In another analysis [41], a combined element of NPI depression, anxiety, and apathy scores exhibited a reduction from baseline in donepezil-treated patients (n = 120). In an RCT [42], the difference between the combination of choline alphoscerate and donepezil vs. donepezil (ASCOMALVA) was statistically significant, for what concerns apathy scores, only in patients with normal frontal assessment battery scores. Overall, in seven RCTs, donepezil demonstrated improvement in apathetic patients.

#### 2.1.2. Galantamine

In a 5-month RCT [43], no deterioration was demonstrated in NPI scores in galantamine-treated patients vs. placebo. On the contrary, in a 3-month multicenter RCT [44], NPI apathy scores in 386 patients did not change significantly with galantamine. In another two multicenter RCTs [45,46], there was an improvement in apathy scores in galantamine-treated patients compared to placebo. Overall, three RCTs demonstrated some clinical benefit in galantamine-treated patients, while one study produced insignificant results. 

#### 2.1.3. Memantine

In a 4-week RCT [47], SCAG and NOSIE apathy scores improved after treatment with memantine. In a following RCT, [48] hobbies/interest BGP subscale also improved in patients who received memantine. On the contrary, insignificant effects were found on apathy scores, following treatment with memantine [49], in patients on a stable donepezil regimen. Overall, two RCTs demonstrated positive effects on apathy scores after memantine treatment, while one RCT produced negative results.

#### 2.1.4. Ginkgo Biloba

In three RCTs, the use of ginkgo biloba was associated with significant improvement in NPI apathy scores [50,51,52].

#### 2.1.5. Methylphenidate

Methylphenidate (20 mg/d) significantly improved apathy scores in an RCT [53]. Similarly, methylphenidate, both in a 5-week cross-over RCT [54] and a 6-week RCT [55], demonstrated significant improvements in apathy scores compared to placebo [56]. In the following 12 weeks, RCT [57], methylphenidate was found to significantly improve apathy score (as assessed by AES-Clinician) vs. placebo. All four RCTs demonstrated benefits in apathy scores in methylphenidate-treated patients.

#### 2.1.6. Modafinil

Modafinil, in an RCT [58], leads to no significant reductions in perceived apathetic symptomatology in patients already receiving an AchEI.

#### 2.1.7. Antidepressants

Antidepressants did not demonstrate a positive effect in apathetic patients with AD dementia [59,60,61,62,63,64]. In an RCT [65], the prescription of citalopram was followed by a non-significant effect on apathy scores. In a following RCT [66], the combination of memantine and citalopram significantly reduced apathy scores vs. memantine with placebo. In an RCT [67], bupropion (150 mg/day) failed to improve apathy vs. placebo, as assessed with the AES-Clinician version.

#### 2.1.8. Atypical Antipsychotics

Atypical antipsychotics have shown favorable therapeutic responses in apathetic AD patients [68,69,70,71]. A re-analysis of the CATIE-AD Study [72], in 421 patients with DSM-IV AD, demonstrated that reduction of NPI apathy score at week 2 was significantly associated with subsequent treatment response with atypical antipsychotics at week 8 (*p* < 0.05). The general use of antipsychotics is not advised in these patients due to serious adverse effects [73]. For a more detailed review of the use of other pharmacological compounds (typical antipsychotics, calcium antagonists, anticonvulsants, etc.) in the treatment of apathy in dementia, see also [27].

#### 2.1.9. Pain Management

In an RCT, 352 patients with dementia and significant behavioral disturbances participated [74]. Patients in the individual daily pain management group exhibited improvements vs. controls in NPI-Nursing Home apathy scores (*p* = 0.017).

#### 2.1.10. THC

In an RCT [75], patients with dementia receiving THC 1.5 mg for 3 weeks failed to exhibit a significant reduction in NPI apathy scores from baseline compared to placebo.

#### 2.1.11. BrainUp-10

In an RCT [76], BrainUp-10^®^ was investigated in mitigating cognitive and behavioral symptoms in patients with AD. Apathy AES scores showed a statistically significant decrease in the group treated with BrainUp-10^®^ at week 4 and at week 12 treatment.

### 2.2. Parkinson’s Disease (PD) and Dementia with Lewy Bodies (DLB)

#### 2.2.1. Memantine

The efficacy and safety of memantine vs. placebo were investigated in an RCT in patients with Parkinson’s disease dementia (PDD) or DLB [77]; insignificant differences in NPI apathy scores were detected. In a 22-week RCT, 25 participants with PDD were randomized to either a placebo or 20 mg/day of memantine [78]. There were no statistically significant differences in NPI apathy score between the memantine-treated group and placebo.

#### 2.2.2. Amantadine

In a 3-month, RCT, parallel-group, wash-out study [79], in 57 amantadine-treated (≥200 mg/d for ≥6 months) dyskinetic PD patients, a significant deterioration in the amantadine group vs. placebo (discontinuing group) was detected, as assessed by the Apathy Inventory (AI) score and scored by the caregiver.

#### 2.2.3. Rivastigmine

In an RCT [80], in 120 patients with LBD, up to 12 mg of rivastigmine daily or a placebo was given for 20 weeks, followed by 3 weeks of rest. NPI Apathy score was reported to improve with rivastigmine treatment. In a multicenter, parallel RCT [81], PD patients with moderate to severe apathy were randomly assigned 1:1 to rivastigmine (transdermal patch of 9.5 mg/day) or placebo for 6 months. Compared with placebo, rivastigmine treatment significantly improved the LARS apathy score (*p* = 0.031).

#### 2.2.4. Rasagiline

In an exploratory post hoc analysis [82], patients with de novo PD taking an antidepressant during the 36-week phase 1 period were randomized to rasagiline (1 or 2 mg/d) or placebo. The pooled rasagiline group revealed a non-significant trend toward reduced worsening in Unified Parkinson’s Disease Rating Scale (UPDRS) item apathy vs. placebo.

#### 2.2.5. Rotigotine

In an RCT [83], PD patients with a total NMSS score ≥ 40 were randomized (2:1) to rotigotine or placebo, titrated over 1–7 weeks to optimal dose (≤8 mg/24 h for patients not receiving levodopa, ≤16 mg/24 h for patients receiving levodopa), maintained for 12 weeks. A numerically greater change (indicating improvement) in the apathy NMSS domain was observed among rotigotine-treated vs. placebo-treated patients: ‘mood/apathy’ (*p* = 0.047; exploratory analyses). The following RCT [84] assessed the 6-month effect of rotigotine vs. placebo on apathy in 48 drug naïve PD patients. Compared to placebo, low-dose rotigotine did not improve LARS apathy scores. In total, 267 patients with PD and unsatisfactory early morning motor impairment were randomized to transdermal patches of rotigotine (2–16 mg/24 h) or placebo [85]. Treatment was titrated to optimal dose over 1–8 weeks, maintained for 4 weeks. Within the NMSS “Mood/apathy” domain, there were significant differences in favor of rotigotine: “lost interest in surroundings” (*p* < 0.0001), “lost interest in doing things” (*p* < 0.0001). A following RCT [86] assessed the efficacy of rotigotine transdermal patch on apathy and motor symptoms in patients with PD-associated apathy (Unified Parkinson’s Disease Rating Scale [UPDRS] I item 4 [motivation] ≥ 2 and patient-rated Apathy Scale [AS] ≥ 14); subjects were randomized 1:1:1 to “low-dose” rotigotine (≤6 mg/24 h for early PD [those not receiving levodopa] or ≤8 mg/24 h for advanced PD [those receiving levodopa]), “high-dose” rotigotine (≤8 mg/24 h for early PD or ≤16 mg/24 h for advanced PD), or placebo, and maintained at optimal/maximal dose for 12 weeks. Rotigotine did not improve PD-associated apathy as rated by the patient. In the RCT by Chung et al. [87], patients with PD were randomized 1:1 to rotigotine or placebo, titrated for ≤7 weeks, and maintained at optimal/maximum dose for 8 weeks. AS scores improved numerically with rotigotine vs. placebo (*p* = 0.0051).

#### 2.2.6. Atomoxetine

In an RCT [88], 55 subjects with PD and an Inventory of Depressive Symptomatology-Clinician (IDS-C) score of 22 were randomized to 8 weeks of atomoxetine or placebo treatment (target dosage = 80 mg/day). AS apathy scores did not differ significantly between the atomoxetine-treated group vs. placebo.

#### 2.2.7. Methylphenidate

In a multicenter, parallel RCT [89], 81 PD patients were randomly assigned to receive methylphenidate (1 mg/kg per day) or placebo for 3 months; the UPDRS and LARS apathy scores in 7 patients receiving methylphenidate improved significantly after 3 months.

#### 2.2.8. Piribedil

A 12-week prospective RCT was conducted in 37 patients with PD presenting with apathy following subthalamic nucleus stimulation. Patients received either piribedil up to 300 mg per day or placebo for 12 weeks [90]. At follow-up evaluation, the Starkstein Apathy Scale score was reduced by 34.6% on piribedil vs. 3.2% on placebo (*p* = 0.015).

#### 2.2.9. IRL752

Patients with PD and associated dementia were randomized to IRL752, a cortical enhancer, or placebo treatment (3:1 ratio) for 28 days [91]. UPDRS item 4 (motivation/initiative) and NPI subdomain apathy/indifference were both improved following IRL752 treatment. The relative change compared to baseline was significant for both severity (*p* = 0.004) and caregiver distress (*p* = 0.029).

#### 2.2.10. Safinamide

Two recent 24-week trials have used safinamide, a dual MAO-I, and glutamatergic transmission modulator up to 100 mg. One [92] showed a trend to significant AES change in ANOVA (*p* = 0.059), while the other [93] was a post hoc analysis of [94] on apathy outcomes from the UPDRS-item 4 (apathy) that showed no significant least squares mean difference (*p* = 0.078) between groups.

**Table 1 brainsci-13-01061-t001:** Quality rates of pharmacological studies using the Physiotherapy Evidence Database rating scale (rated items only are displayed) and the Oxford Center of Evidence-Based Medicine criteria.

Ref.	Compound/Apathy Measure	PEDro	PEDro	PEDro	PEDro	PEDro	PEDro	PEDro	PEDro	PEDro	PEDro	PEDro	PEDro	OCEBM
		Random Group Allocation	Allocation Concealed	Baseline Group Similarity	Blinding of All Subjects	Blinding of All Therapists	Blinding of All Assessors of At Least one Key Outcome	Less than 15% Dropouts	Intention to Treat Analysis of At Least One Key Outcome	Between-Group Statistical Comparisons Reported for At Least One Key Outcome	Point Measurements and Measurements of Variability Provided for At least One Key Outcome	Total Yes	Quality	
Alzheimer’s Disease														
Tariot et al. [35]	Donepezil/NPI	Y	Y	Y	Y	Y	Y	N	Y	Y	Y	9	High	B
Feldman et al. [36]	Donepezil/NPI	Y	Y	Y	Y	Y	Y	N	Y	Y	Y	9	High	B
Gauthier et al. [37]	Donepezil/NPI	Y	Y	Y	Y	Y	Y	N	Y	Y	Y	9	High	B
Feldman et al. [38]	Donepezil/NPI	Y	Y	Y	Y	Y	Y	N	Y	Y	Y	9	High	B
Holmes et al. [39]	Donepezil/NPI	Y	Y	Y	Y	Y	Y	N	Y	Y	Y	9	High	B
Seltzer et al. [40]	Donepezil/AES	Y	N	N	Y	Y	Y	N	Y	N	N	5	Moderate	B
Cummings et al. [41]	Donepezil/NPI	Y	N	Y	Y	Y	Y	N	Y	Y	Y	8	High	B
Rea et al. [42]	Donepezil+choline alphoscerate/NPI	Y	Y	Y	Y	Y	Y	Y	N	Y	Y	9	High	B
Tariot et al. [43]	Galantamine/NPI	Y	Y	Y	Y	Y	Y	N	Y	Y	Y	9	High	B
Rockwood et al. [44]	Galantamine/NPI	Y	Y	Y	Y	Y	Y	N	Y	Y	Y	9	High	B
Erkinjutti et al. [45]	Galantamine/NPI	Y	Y	Y	Y	Y	Y	N	Y	Y	Y	9	High	B
Cummings et al. [46]	Galantamine/NPI	Y	Y	Y	Y	Y	Y	N	Y	Y	Y	9	High	B
Winblad and Poritis [48]	Memantine/CGI-C	Y	N	Y	Y	Y	Y	Y	Y	Y	Y	9	High	B
Cummings et al. [49]	Memantine/NPI	Y	Y	Y	Y	Y	Y	N	Y	Y	Y	9	High	B
Scripnikov et al. [50]	Ginkgo Biloba/NPI	N	N	Y	Y	Y	Y	N	N	Y	Y	7	High	B
Bachinskaya et al. [51]	Ginkgo Biloba/NPI	Y	N	Y	Y	Y	Y	N	Y	Y	Y	8	High	B
Ihl et al. [52]	Ginkgo Biloba/NPI	Y	N	Y	Y	Y	Y	Y	Y	Y	Y	9	High	B
Kaplitz et al. [53]	Methylphenydate/NOSIE	Y	N	N	Y	Y	Y	N	N	Y	Y	6	Moderate	C
Hermann et al. [54]	Methylphenydate/AES	Y	N	N	Y	Y	Y	N	N	Y	Y	6	Moderate	C
Rosenberg et al. [55]	Methylphenydate/AES, NPI	Y	Y	Y	Y	Y	Y	Y	Y	Y	Y	10	High	A
Padala et al. [57]	Methylphenydate/AES-C	Y	Y	N	Y	Y	Y	Y	N	Y	Y	8	High	B
Frakey et al. [58]	Modafinil/FSBS	Y	N	Y	Y	Y	Y	Y	N	Y	Y	8	High	B
Zhou et al. [66]	Citalopram/NPI	Y	N	Y	Y	N	N	N	N	Y	Y	5	Moderate	B
Maier et al. [67]	Bupropion/AES-C	Y	N	Y	Y	Y	Y	N	Y	Y	Y	8	High	B
Nagata et al. [72]	Atypical antipsychotics/NPI	Y	N	Y	Y	N	N	Y	Y	Y	Y	7	High	B
Husebo et al. [74]	Pain treatment (paracetamol, morphine XR, buprenorphine or pregabaline)/NPI-NH	Y	Y	N	Y	Y	Y	N	N	Y	Y	7	High	B
van den Elsen et al. [75]	Tetrahydrocannabinol/NPI	Y	Y	Y	Y	Y	Y	Y	Y	N	N	8	High	B
Guzman-Martinez et al. [76]	BrainUp-10^®^/AES	Y	Y	Y	Y	Y	Y	Y	N	Y	Y	9	High	B
**Parkinson’s Disease**														
Emre et al. [77]	Memantine/NPI	Y	Y	Y	Y	Y	Y	N	Y	Y	Y	9	High	B
Leroi et al. [78]	Memantine/NPI	Y	N	N	Y	Y	N	Y	N	Y	Y	6	Moderate	B
Ory-Magne et al. [79]	Amantadine/AI	Y	Y	Y	Y	Y	Y	Y	Y	Y	Y	10	High	A
Devos et al. [81]	Rivastigmine/LARS	Y	Y	Y	Y	Y	Y	Y	Y	Y	Y	10	High	A
Smith et al. [82]	Rasagiline/UPDRS	Y	N	Y	Y	Y	Y	N	N	Y	Y	7	High	B
Antonini et al. [83]	Rotigotine/NMSS	Υ	Ν	Υ	Υ	Υ	Υ	Ν	Υ	Υ	Υ	8	High	B
Castriotto et al. [84]	Rotigotine/LARS	Y	Y	Y	Y	Y	Y	Y	Y	Y	Y	10	High	A
Chaudhuri et al. [85]	Rotigotine/NMSS	Y	N	Y	Y	Y	Y	Y	N	Y	Y	8	High	B
Hauser et al. [86]	Rotigotine/UPDRS	Y	Y	Y	Y	Y	Y	N	Y	Y	Y	9	High	B
Chung et al. [87]	Rotigotine/AES	Y	N	Y	Y	Y	Y	N	Y	Y	Y	8	High	B
Weintraub et al. [88]	Atomoxetine/AES	Y	N	Y	Y	Y	Y	Y	Y	Y	Y	9	High	B
Moreau et al. [89]	Methylphenydate/UPDRS, LARS	Y	Y	Y	Y	Y	Y	Y	Y	Y	Y	10	High	A
Thobois et al. [90]	Piribedil/AES	Y	Y	Y	Y	Y	Y	Y	Y	Y	Y	10	High	A
Sveningson et al. [91]	IRL752/UPDRS, NPI	Y	N	N	Y	Y	Y	Y	Y	Y	Y	8	High	B
Hattori et al. [94]	Safinamide/UPDRS	Y	N	Y	Y	Y	Y	Y	N	Y	Y	8	High	B
Kulisevsky et al. [92]	Safinamide/NPI, AES	Y	N	Y	Y	Y	Y	Y	Y	Y	Y	9	High	B
Dementia with Lewy Bodies														
McKeith et al. [80]	Rivastigmine/NPI	Y	Y	Y	Y	Y	Y	N	Y	Y	Y	9	High	B
Frontotemporal Dementia														
Jesso et al. [95]	Oxytocin/NPI	Υ	N	N	Υ	Υ	Υ	Y	N	Y	Y	7	High	B
Finger et al. [96]	Oxytocin/NPI, FBI	Y	N	Y	Y	Y	Y	Y	N	Y	Y	8	High	B
Huey et al. [97]	Dextroamphetamine vs. Quetiapine/NPI	Y	N	Y	Y	Y	Y	Y	N	Y	N	7	High	B
Callegari et al. [98]	Melatonin/AES-C, NPI	Y	N	Y	Y	Y	Y	Y	N	Y	Y	9	High	B
Huntington’s Disease														
Gelderblom et al. [99]	Bupropion/AES, NPI, UPDRS	Y	Y	Y	Y	Y	Y	Y	Y	Y	Y	10	High	A

NPI-NH: Neuropsychiatric Inventory-Nursing Home. AES-C: Apathy Evaluation Scale-Clinician. CGI-C: Clinical Global Impression of Change inventory. NOSIE: Nurses Observation Scale for Inpatient Evaluation. FSBS: Frontal Systems Behavior Scale. AI: Apathy Inventory. LARS: Lille Apathy Rating Scale. UPDRS: Unified Parkinson’s Disease Rating Scale. NMSS: Non-Motor Symptoms Scale in Parkinson’s disease. FBI: Frontal Behavioral Inventory. OCEBM: Oxford Center of Evidence-Based Medicine; PEDro: Physiotherapy Evidence Database; Y: Yes; N: No.

### 2.3. Frontotemporal Dementia (FTD)

#### 2.3.1. Oxytocin

In an RCT with a cross-over design [95], 20 patients with behavioral variant FTD (bvFTD) received one dose of 24 IU of intranasal oxytocin or a placebo. Caregivers completed validated behavioral ratings at 8 h and 1 week following drug administrations. Post hoc exploratory examination of subitem NPI scores did not reveal significant differences between individual NPI apathy scores for Day 1 of oxytocin vs. placebo. In a parallel-group RCT [96], a dose-escalation design was used to test 3 clinically feasible doses of intranasal oxytocin (24, 48, or 72 IU) administered twice daily for 1 week to 23 patients with bvFTD or semantic dementia. Possible trends of improvement for the NPI apathy (FBI) apathy domains were found.

#### 2.3.2. Dextroamphetamine

In a cross-over RCT, the authors [97] contrasted the effects of dextroamphetamine and quetiapine in 8 patients with bvFTD. The NPI subscales that decreased the most on dextroamphetamine were apathy (2.8 points) and disinhibition (2.4 points).

#### 2.3.3. Agomelatine

In a double-blind procedure, 24 non-depressed patients with a diagnosis of bvFTD patients were randomized, using a cross-over design, to receive either agomelatine 50 mg/day or sustained release melatonin 10 mg/day for 10 weeks [98]. At the end of the follow-up period, subjects receiving melatonin switched to agomelatine for the following 10 weeks. Agomelatine, but not melatonin, was associated with a significant reduction of AES-Clinician score in FTD subjects and of caregiver distress due to patients’ apathy (NPI-A-distress score).

### 2.4. Huntington’s Disease

#### Bupropion

In a multicenter, cross-over RCT, individuals with HD and clinical signs of apathy according to the Structured Clinical Interview for Apathy-Dementia (SCIA-D), but not depression (n = 40) were randomized to receive either bupropion 150/300 mg or placebo daily for 10 weeks [99]. There was no statistically significant difference between treatment groups for all clinical primary (AES, AES-I) and secondary outcome variables (AES-Clinician, AES-S (self), NPI, and UHDRS apathy scores). Study participation, irrespective of the intervention, lessened symptoms of apathy, according to the informant and the clinical investigator.

## 3. Discussion

### 3.1. Principal Findings

We attempted to systematically review pharmacological treatments for apathy in aging neurocognitive disorders, including combined treatments.

Alzheimer’s Disease: In most of the RCTs in AD, apathy was considered as a secondary outcome measure. AchEIs [100,101], gingko biloba [50,51,52], and methylphenidate have been shown to be effective; memantine was not beneficial [101]. More research is needed in this domain. Although beneficial, prolonged use of atypical antipsychotics is not advised [73], while antidepressants [59,60,61,62,63,64,65] did not have a positive outcome.

Parkinson’s Disease: For concerned patients with PD, rivastigmine [80,81] was found to be beneficial in two RCTs, cognitive enhancer IRL752 [91] in one RCT and piribedil [90] in one RCT; while rotigotine [83,84,85,86,87] was found efficacious in some RCTs in patients with PD but not in all of them.

Frontotemporal Dementia: Regarding patients with FTD, agomelatine [98] was found to be beneficial for apathy in one RCT.

### 3.2. Strengths and Weaknesses

#### 3.2.1. Appraisal of Methodological Quality of the Review

We evaluated the quality of the reported evidence using two published semi-quantitative methods. In most of the RCTs included, apathy was not a primary outcome measure; the NPI-apathy score was used, which may not be consistent with more specific apathy scales [55]. Furthermore, effect sizes and parameters were not reported in a reliable way in all RCTs, and this would make the organization of a meta-analysis very difficult. Several limitations apply to this review. Any conclusions drawn rely on the quality of the included studies, while an unknown number of studies that have combined treatments might have been excluded by the review design in older reviews. The studies included in this review were methodologically heterogeneous and certainly, had small sample sizes. It should be noted that age and gender may have an effect on pharmacological treatment [102]. Studies were not so prone to publication bias, apathy being the secondary outcome in more cases; however, selective reporting cannot be excluded.

#### 3.2.2. Preceding Pharmacological Reviews

There are only a few reviews that specifically have focused on apathy outcomes in aging neurocognitive disorders following pharmacological interventions [22,101,103,104,105,106,107,108,109,110,111,112]. In a study [106] based on 2 RCTs (35,106], it was suggested that donepezil may improve apathy scores in AD. In a meta-analysis [113] of 3 RCTs [43,44], treatment with galantamine resulted in non-significant improvements in apathy scores. A study [114] analyzed the data of 2 RCTs [49,115] and found that memantine-treated participants exhibited an improvement in NPI apathy scores. However, in an analysis [116] of RCTs [115,117,118,119,120,121], patients treated with memantine demonstrated insignificant results in apathy scores vs. placebo. In a recent study [107], patients with prodromal dementia exhibited mild reductions in apathy scores after treatment with galantamine plus risperidone. Furthermore, a recent meta-analysis [108] demonstrated that methylphenidate may be beneficial for apathy in people with AD; however, the authors underline that this finding is associated with low-quality evidence. It is worth noting that the recently published report of the ADMET 2 RCT study [122] revealed a significantly larger 6-month difference in NPI apathy score and ADCS-CGIC apathy score in the methylphenidate group vs. placebo group. Interestingly, the difference in NPI apathy decrease was achieved sooner in the methylphenidate group than in the placebo group.

In a review [22] on pharmacological treatments for apathy in various dementia types, it was suggested that there is no convincing proof for any of the drugs. Similarly, in a meta-analysis [109], the authors failed to find a significant treatment effect for apathy in favor of any drug.

Rectorova [110], in her review of the treatment of PD, has reached the conclusion that rivastigmine and piribedil are beneficial for apathy in PD. In comparison, Mele et al. [111] concluded that there is limited evidence on gold standard treatment for apathy in PD. Wang et al. [112], in a meta-analysis, have concluded that rotigotine transdermal patches effectively improved apathy in patients with PD.

### 3.3. General Implications for Future Research

In general, apathy in aging neurocognitive disorders is not very well investigated. While research in apathy treatment in AD is ongoing, studies in PD, where apathy is highly prevalent, are lacking. The aggregated evidence from high-quality studies in our review showed that treatment-favoring results come from methylphenidate [108], donepezil, and gingko biloba in AD, and less favoring results from galantamine, memantine in AD and rotigotine in PD (see Table 2). The literature on studies that have used other compounds is limited, and a lack of studies in the pharmacological treatment of apathy in specific neurodegenerative processes such as DLB and HD was observed. More studies are needed to further test current results, other compounds (such as Oxytocin in FTLD [95,96]), and different neurodegenerative conditions.

Heterogeneity in pathology and disease stage, as well as inherent diagnostic weaknesses in aging neurodegenerative disorders, make the interpretation of the findings difficult and cautious. For example, because the exact underlying pathophysiological mechanisms of the different neurodegenerative disease processes are not known, it is problematic to attribute treatment’s effects to a specific molecular mechanism in a given disease stage. On the other hand, apathy in AD and PD responds to catecholaminergic and cholinergic enhancement, which highlights brain systems to target with treatments. Future studies are encouraged to be conducted in well-characterized groups in terms of the type and stage of neurodegeneration. Moreover, there is a need for a unified framework for apathy assessment and treatment outcome evaluation by adopting validated structural measures [15]. This framework may be different with respect to the degenerative condition.

It is not known how apathy responds to a given treatment in different disease stages and across neurodegenerative processes. State, contextual, and other issues may influence treatment response, and more studies are needed in this direction. The design of the studies should promote individualized therapies, environmental and cultural considerations, and psychoeducation. Furthermore, there are issues regarding not only the design of studies (e.g., randomization, blinding) but also the implementation (the natural variation of the interventionists, environmental considerations, and contextual issues (NHs, day care centers)); all these parameters could affect outcomes. Consequently, it is not clear whether interventions have different effects when administered in different care settings. Furthermore, the diversity of the outcome measures is an important issue. Interestingly, in two reviews [123,124], the DAIR, the AES, and the NPI were found to be among the best quality apathy scales. While NPI was the instrument most frequently used to measure apathy, it has been used in different neurodegenerative states, and thus it is not known how well NPI captures the apathy symptoms in different neurodegenerative states and evaluates the outcome. This pertains to a wider issue concerning the scales and their sensitivity to the type and stage of degeneration, and this implies further research in this domain.

Furthermore, appropriately modifying and further evolving existing scales (e.g., the Lille Apathy Rating scale-LARS) [125] is of interest. Because most of the studies aimed at the effectiveness of the intervention, it is critical to incorporate more efficacy studies criteria in order to add internal validity to generalizability. Recommendations [126,127] on the design of clinical trials on apathy have recently been published. Furthermore, it should be underlined that endorsement of the (CONSORT) statement could make trials more complete. Additionally, it would be interesting to investigate health outcomes and further individual implications by interventions administered for longer periods against the cost.

### 3.4. Conclusions

A combination of appropriate non-pharmacological, medical, and pharmacological therapeutic interventions in patients with aging neurocognitive disorders could enable clinicians to choose optimal treatment plans [128]. Furthermore, prevention of apathy and MCI [129] should be vigorously pursued by endorsing exercise, and cognitive stimulating activities.

## Figures and Tables

**Figure 1 brainsci-13-01061-f001:**
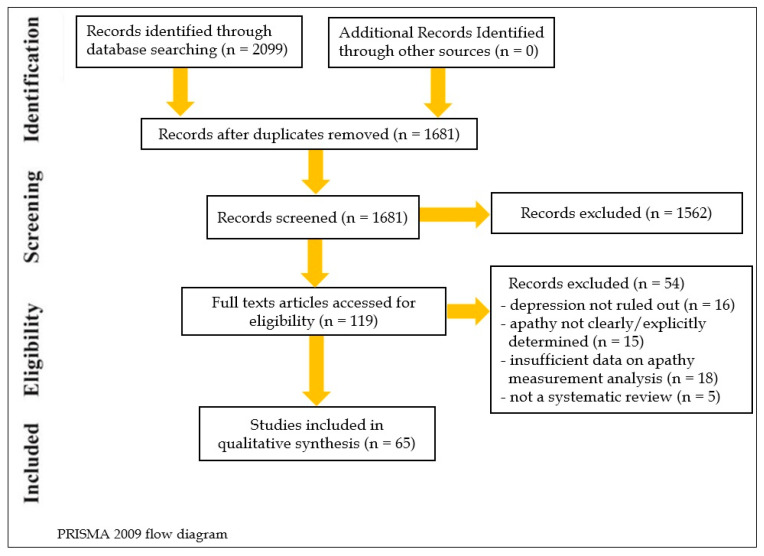
PRISMA 2009 Flow Diagram.

**Table 2 brainsci-13-01061-t002:** Summary of Medications and Their Efficacy in the Treatment of Apathy in RCTs included in this Review.

Medication	Is It Beneficial?	Evidence Supporting RCTs/Total RCTs	Quality of Evidence
Alzheimer’s Disease			
Donepezil	Yes	6/7	⋆⋆⋆⋆
Galantamine	Yes	3/4	⋆⋆⋆⋆
Memantine	Yes	2/3	⋆⋆
Ginkgo Biloba	Yes	3/3	⋆⋆⋆⋆
Methylphenidate	Yes	4/4	⋆⋆⋆⋆
Modafinil	No	0/1	--
Antidepressants	No	1/7	--
Atypical antipsychotics	Not generally advised		
Pain management	Yes	1/1	⋆
THC	No	0/1	--
BrainUp-10	Yes	1/1	⋆
Parkinson’s Disease			
Memantine	No	0/2	--
Amantadine	No	0/1	--
Rivastigmine	Yes	1/1	⋆
Rasagiline	No	0/1	--
Rotigotine	Yes	3/5	⋆⋆
Atomoxetine	No	0/1	--
Methylphenidate	Yes	1/1	⋆
Piribedil	Yes	1/1	⋆
IRL752	Yes	1/1	⋆
Safinamide	No	0/2	--
Dementia with Lewy bodies			
Rivastigmine	Yes	1/1	⋆
Frontotemporal Dementia			
Oxytocin	No	0/2	--
Dextroamphetamine	Yes	1/1	⋆
Agomelatine	Yes	1/1	⋆
Huntington’s Disease			
Bupropion	No	0/1	--

⋆⋆⋆⋆ Multiple peer-reviewed RCTs, replicating findings, effects are meaningful (moderate to large); ⋆⋆ Mixed empirical evidence, some positive some negative, or effect sizes are variable or moderated by intervening or unknown variables; ⋆ Only one or two studies, but clinicians report good results and come from more than one setting, hospital, treatment center; --No positive supporting evidence.

## Data Availability

The dataset used and/or analyzed during this study is available from the corresponding author upon reasonable request.

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
