# Peer review of "A Systematic Review of Pharmacological Interventions for Apathy in Aging Neurocognitive Disorders"

_brainsci, 2023, doi:10.3390/brainsci13071061_

Round 1

Reviewer 1 Report

This paper does exactly what its title promises:  It provides a systematic review of pharmacological interventions for apathy – a state that may arise in aging neurocognitive disorders. The review is nicely framed in the introduction, and fits well with the purpose of the special issue.  The systematic review was very well thought-out and was conducted very rigorously.  The results of the review are presented in a well-organized manner (with each neurocognitive disorder reviewed in turn).  The writing and language-use are very clear and coherent.  The paper as a whole is outstanding, and worthy of publication as is.  I think it is an excellent contribution to the literature -- and it was a pleasure to read. (Thank you!)

As a footnote -- if you, as authors, have an opportunity and inclination to make some minor optional revisions, I have attached a file that contains 3 suggestions that you can consider.  i want to emphasize that these are intended to be entirely optional, and do not need to be added/changed in the paper from my perspective.

1.     

Author Response

We thank you for your comments, please see our response to your queries in the pdf file below.

Reviewer 2 Report

See file for comments

There are a number of grammatical errors but these are minor and can be corrected with proof reading

Author Response

We thank you for your comments, we have corrected the grammatical errors with proof reading.

Reviewer 3 Report

The authors develop this study with the aim of conducting a systematic review to investigate the effects of pharmacological treatment for neurodegenerative diseases on apathy. To do this, they first introduce the concept of apathy and its definition, then present the methodologies with which they conducted their research and finally show the results obtained. The topic is very interesting and has a lot of value in the field of neuroscience, since over the years scientific literature has increasingly demonstrated the importance of the role of apathy in neurodegenerative diseases and immediate intervention on it is desirable in order to slow down or reduce the onset of symptoms.

However, I ask the authors to consider the following comments to improve and implement the quality and robustness of the manuscript.

Comment 1: The quality of English needs to be improved. Sometimes the complexity in the construction of the sentences or the repetitiveness of some linguistic forms make reading not very smooth and fluid. Authors are encouraged to do a minor review of the manuscript. The use of appropriate terminology is critical to the context. Authors are encouraged to rephrase some sentences and review the use of some terms.

Comment 2: The introduction is well centred on the topic of apathy. However, it is believed that the bibliography used to discuss the topic is insufficient. The authors briefly introduce the theme, relying on a few bibliographic references. It is believed that perhaps it would be more appropriate to deepen the definition of apathy and its importance in the field of neurodegenerative diseases. Furthermore, the symptom of apathy can have different manifestations and meanings in different pathologies. It is not clear how and according to what criteria the authors decided to include the treatment of this symptom in various pathologies, even if neurodegenerative. The authors are therefore asked to explain how they intend to consider the differences related to the various pathologies taken into consideration.

Comment 3: Paragraph numbering appears inconsistent with text. For example, the introductory paragraph is treated as paragraph 1 and the search strategy paragraphs as its subparagraphs. Authors are requested to separate the introduction from the paragraphs on methodology and proceed with appropriate numbering. Furthermore, it is not clear whether the title of paragraph 2 is the title of the paragraph or the title of table 2. The authors are requested to separate the indication of the table from the title of the paragraph in order to favour a more immediate understanding of the structure of the paper.

Comment 4: In the Search strategy and study selection paragraph, the criteria used for the inclusion and exclusion of the studies selected for their research are not clearly and immediately explained. Authors are requested to review the organization of the paragraph to make it more systematic and smoother.

Comment 5: The numbering of the tables needs to be reviewed. In fact, the authors indicate a table 2, but there is no table numbered 1 in the paper. The authors are requested to correct the numbering of the table.

Comment 6: The discussion paragraph should also be numbered in a systematic and orderly manner. If the "Strengths and Weaknesses" subparagraph refers to the Discussion paragraph it should be numbered as 3.2. Otherwise, if it is a separate paragraph, it must be indicated in capital letters and/or in bold, as was done for all the other paragraphs. Furthermore, if this paragraph were on its own, it would not be necessary to number paragraph 3.1. (Principal findings) because it would be the only paragraph of the discussion. In paragraph 4 "Strengths and Weaknesses" there appear to be sub-paragraphs, indicated by what appear to be titles within the text (Appraisal of methodological quality of the review, line 312; Several limitations apply to this review, line 318; Relation to other pharmacological reviews, line 325). The authors are requested to revise and reorganize the paragraphs in an appropriate and coherent manner with the sense of the text.

Comment 7: As regards Table 2, it is not clear how the Physiotherapy Evidence Database rating scale had an impact on the selection of articles and data processing. Furthermore, the authors refer to a bibliographic entry for explanations on this topic. However, it is believed that the authors can better explain the processes used and their goals and intents. In fact, it is assumed that the paper can be understood for its contents without the need to consult further papers to understand the meaning and purposes of the authors. The authors are therefore invited to clarify more precisely how they used certain data and what impact they had on the study. Once these topics have been explained in more detail, please also make the table more intuitive and clearer.

Comment 8: The authors are asked to explain why they decided to include studies only up to 2021 in the research. This could be considered as a considerable limitation since more than a year of studies are excluded from the research, thus making the work and the results less current.

The quality of English needs to be improved. Sometimes the complexity in the construction of the sentences or the repetitiveness of some linguistic forms make reading not very smooth and fluid. Authors are encouraged to do a minor review of the manuscript. The use of appropriate terminology is critical to the context. Authors are encouraged to rephrase some sentences and review the use of some terms.

Author Response

We thank you for your comment please see our response to your queries in the pdf file below.

Reviewer 4 Report

Theleritis et a. performed a systematic review of pharmacological interventions for apathy in various neurodegenerative disorders. The topic is of interest and the review well organized. Minor concerns should be addressed:

-Introduction: this section is quite short and lacks of relevant study (see for example Spalletta et a., 2013). Please expand it.

- Methods: search strategy is unclear. In line 78, authors described a long list of additional search items. Were they all included in the study selection process? Please clarify. Moreover, the authors should provide an extensive table with the studies included in the review, describing the relevant variables that they addressed for results analysis.

- Please modify the format of Table 2;

- Please spell out the acronym "MCI". 

Author Response

(The authors gave the same response as above.)
